# In-Context Learning Creates Task Vectors

**Roee Hendel**
Tel Aviv University
roee.hendel@mail.tau.ac.il

**Mor Geva**
Google DeepMind
pipek@google.com

**Amir Globerson**
Tel Aviv University, Google
gamir@tauex.tau.ac.il

## Abstract

In-context learning (ICL) in Large Language Models (LLMs) has emerged as a powerful new learning paradigm. However, its underlying mechanism is still not well understood. In particular, it is challenging to map it to the "standard" machine learning framework, where one uses a training set $S$ to find a best-fitting function $f(x)$ in some hypothesis class. Here we make progress on this problem by showing that the functions learned by ICL often have a very simple structure: they correspond to the transformer LLM whose only inputs are the query $x$ and a single "task vector" calculated from the training set. Thus, ICL can be seen as compressing $S$ into a single task vector $\boldsymbol{\theta}(S)$ and then using this task vector to modulate the transformer to produce the output. We support the above claim via comprehensive experiments across a range of models and tasks.[1]

## 1 Introduction

Large language models have improved dramatically over the last several years. One striking property of these models is that they can learn new rules from very few demonstrations. For instance, a model can be prompted with the input *"Apple → Red, Lime → Green, Corn →"* and produce the output *"Yellow"*. The model has thus learned a mapping based on just two examples, which it can apply correctly to new examples. This capability, referred to as In-Context Learning (ICL), has been used extensively, yielding impressive empirical results ([Brown et al., 2020](); [Liu et al., 2023](); [Dong et al., 2022]()).

Given this success, it is natural to ask what is the underlying mechanism behind ICL. Namely, how does the model internally use the demonstrations $S$ and the query $x$ to produce the required output? Here we approach this question by utilizing the

---

[1]We release our code at https://github.com/roeehendel/icl_task_vectors.

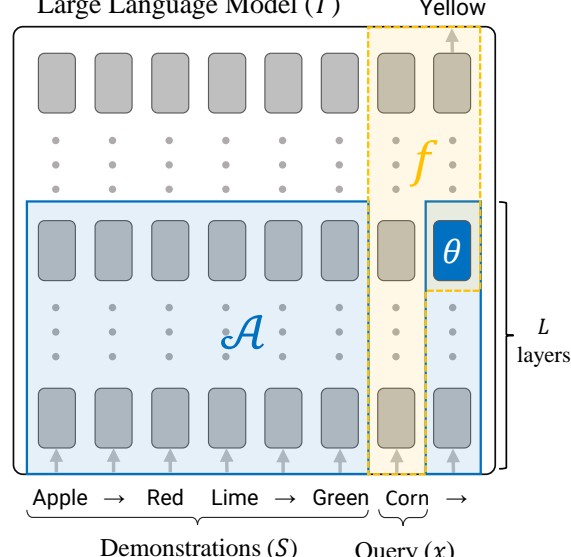

Figure 1: **ICL as learning in a Hypothesis Class.** In ICL, one provides an LLM with a prompt including demonstrations $S$ of some task, and a query $x$. The model generates the output for $x$ (here "Yellow"). We show that the underlying process can be broken down into two parts: $\mathcal{A}$, a "learning algorithm" (marked in blue), computes a query-agnostic vector $\boldsymbol{\theta}(S)$, which we view as a parameter of a function in a hypothesis class. The second part, denoted by $f$ and marked in yellow, is the application of the rule defined by $\boldsymbol{\theta}$ on the query $x$, without direct dependence on $S$.

concept of a hypothesis class from statistical learning theory ([Shalev-Shwartz and Ben-David, 2014]()). In the learning-theoretic formulation, one typically considers a hypothesis class $\mathcal{H}$, where every element of $\mathcal{H}$ is a function $h(x; \boldsymbol{\theta})$, operating on the input $x$, and specified by a parameter vector $\boldsymbol{\theta}$. For example, if $x \in \mathbb{R}^d$ then the class $\mathcal{H}$ could be the set of linear classifiers, defined by a coefficient vector $\boldsymbol{\theta}$ as $h(x; \boldsymbol{\theta}) = \boldsymbol{\theta} \cdot x$. Learning algorithms seek an element $h \in \mathcal{H}$ that fits the training set well. This is known as Empirical Risk Minimization.

It is unclear whether ICL operates in such a way because the prediction is performed via $T([S, x])$, where $T$ is typically an auto-regressive transformer

and $[S, x]$ is a concatenation of the tokens in $S$ and $x$. Thus, in the general case, it can be an arbitrary function that operates on $S$ and $x$ to produce the output. This can include "non-parametric" methods such as nearest-neighbor. Recent work has begun to explore this question. For example, it was shown that when training a transformer from scratch to perform linear regression in context, the emerging learning algorithm is similar to Stochastic Gradient Descent (Akyürek et al., 2022; von Oswald et al., 2022). However, for LLMs performing more complex natural language tasks, it is not at all clear what the hypothesis space may be.

In this work, we show that on a wide range of tasks, ICL in LLMs can be viewed as working on a very natural hypothesis space. We argue that, given a training set $S$, the transformer maps it into a "task vector" $\boldsymbol{\theta}(S)$ that essentially represents the mapping/rule described in $S$.[2] Namely, given the transformer $T$ and a vector $\boldsymbol{\theta}$, we can construct a new function $f(x; \boldsymbol{\theta})$ that implements the task. The function $f$ is very similar to the original transformer applied to $x$ *without* demonstrations but instead modulated by $\boldsymbol{\theta}$ (see Fig. 2).

Our view is also related to soft prompts (Lester et al., 2021), since both approaches modulate the function of the transformer towards a particular task. However, in ICL, task vectors are calculated in the forward pass rather than being fine-tuned.

Our contributions include proposing a hypothesis-class based mechanistic view of ICL, and conducting experiments to validate our view on a range of publicly available LLMs and a diverse set of tasks. Our results further the understanding of ICL and may have practical implications for the efficient adaptation of LLMs to perform specific tasks.

## 2 A Hypothesis Class View of ICL

Motivated by the hypothesis class view of learning theory, our goal is to understand if ICL maps the set of demonstrations $S$ to a function on the query $x$ and how this mapping occurs. Specifically, we seek to see if ICL converts $S$ into $\boldsymbol{\theta}$ - the "parameters" of a function within a certain hypothesis space. Our empirical findings suggest this view is applicable, shedding light on the structure of the hypothesis space on which ICL can be viewed to operate.

---

[2]The term "task vector" was coined by Ilharco et al. (2023) for directions in weight space that correspond to a particular task. Although our vectors are in "activations space" they share a similar motivation and thus we overload the term.

### 2.1 Theoretical Framework

We use $T$ to denote a decoder-only transformer LLM, $S$ to denote the set of demonstrations (i.e. training examples) used as input to ICL, and $x$ to denote the query that ICL is asked to provide an output for. We use $T([S, x])$ to denote the output of ICL on the concatenation of $S$ and $x$.

To demonstrate that ICL operates within a hypothesis space, we aim to show that its underlying mechanism can be broken down into two parts:

- A **"Learning Algorithm"** (denoted by $\mathcal{A}$) that maps $S$ into a "task vector" $\boldsymbol{\theta}$, independent of the query $x$. Given that attention layers can access both $S$ and $x$, this independence is not trivial.
- A **"Rule Application"** (denoted by $f$) which maps the query $x$ to the output, based on $\boldsymbol{\theta} \equiv \mathcal{A}(S)$, without direct dependence on $S$. Again, this independence is not trivial.

Thus, we consider the following mapping from a set of demonstrations and a query to the predicted output: $T([S, x]) = f(x; \mathcal{A}(S))$.

If we can break down the forward pass of the LLM into the above two components, we can view ICL as operating on the following hypothesis class: $\mathcal{H} = \{f(\cdot; \boldsymbol{\theta}) \mid \boldsymbol{\theta}\}$. In the next section we propose an implementation of such a class.

### 2.2 A Proposed Hypothesis Class

There are many possible realizations of the above framework, that correspond to different choices of $\mathcal{A}$ and $f$. We next describe the realization we focus on, which naturally follows from the transformer architecture. We consider an ICL setting as in Fig. 1, where the input ends with a query $x$ (i.e., Corn) followed by an "→" symbol. As mentioned above, we view learning as composed of two steps: calculating a parameter vector $\boldsymbol{\theta}$ based on the training sample $S$, and applying the rule defined by this parameter vector to the query $x$. A presumably simple way for a transformer to do this is for the first $L$ layers of the → representations to calculate $\boldsymbol{\theta}$ and then for the remaining layers to take $\boldsymbol{\theta}$ and $x$ as input and produce an output. See Fig. 1. Recall that $S$ and $x$ are accessible to the transformer at any layer, presenting a challenge with our view.

In the following sections, we address this challenge and present experiments validating our view. Namely, we show that we can isolate our proposed $\mathcal{A}$ and $f$ in the forward pass of LLMs performing ICL. We also show that the $\boldsymbol{\theta}$ vectors are interpretable and correspond to learned tasks.

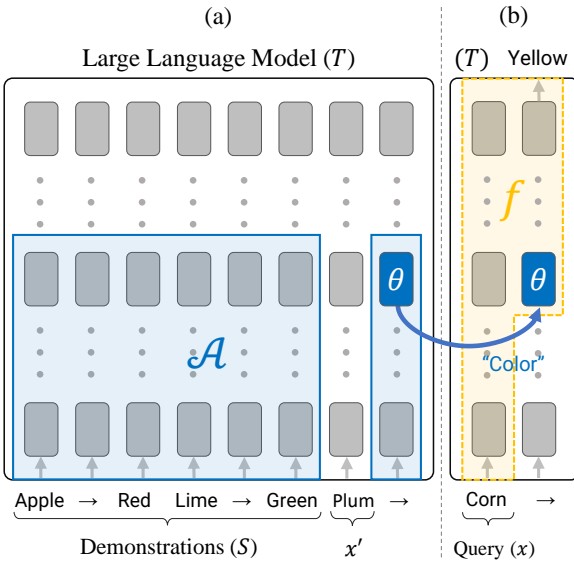

(a) Large Language Model ($T$)

(b) ($T$) Yellow

$\theta$

$f$

$\theta$

$\mathcal{A}$

"Color"

Apple → Red   Lime → Green   Plum →   Corn →

Demonstrations ($S$)   $x'$   Query ($x$)

Figure 2: **Separating $\mathcal{A}$ and $f$.** To make $\boldsymbol{\theta}$ independent of the query $x$, we use a dummy query ($x' = $ Plum) and use the representation of → at the $L^{th}$ layer as $\boldsymbol{\theta}$. The vector $\boldsymbol{\theta}$ is then patched at the same layer during a forward pass of a transformer that only takes $x$ and → as input, to prevent the direct dependence of $f$ on $S$.

## 3 Validity of the Hypothesis Class View

We first show that separating the forward pass into the two distinct components $\mathcal{A}$ and $f$, defined in §2.2, maintains the high accuracy of ICL.

### 3.1 Separating $\mathcal{A}$ and $f$

We face some challenges in a regular forward pass: first, the initial $L$ layers that correspond to $\mathcal{A}$, updating the representations of → to create $\boldsymbol{\theta}$, can attend to the query $x$. Thus, they may depend on $x$, creating an unwanted dependence of $\boldsymbol{\theta}$ on $x$. Second, the remaining layers that correspond to $f$, may directly access $S$, instead of using only $x$ and $\boldsymbol{\theta}$.

We propose the following procedure to tackle these challenges: to solve the first problem, we introduce a "dummy query" $x'$ and calculate the representations of → using that query. We use the representation of → after the first $L$ layers, calculated using $x'$, as the vector $\boldsymbol{\theta}$ (as demonstrated on the left side of Fig. 2). An alternative was to block attention to $x$, but it led to poor performance. To solve the second problem of calculating $f(x, \boldsymbol{\theta})$ without allowing direct dependence on $S$, we perform a forward pass of the transformer only on $x$ and →,[3] and "patch" the $\boldsymbol{\theta}$ we previously extracted at the $L$th layer of the → (right side of Fig. 2).[4]

---

[3]Ignoring positional embeddings, this is equivalent to blocking the attention to $S$ in these layers.

[4]Note that the second token can actually be anything, because it is overridden by patching. We use → for simplicity.

| Category | Task | Example |
|---|---|---|
| Algorithmic | Next letter | a → b |
| | List first | a,b,c → a |
| | List last | a,b,c → c |
| | To uppercase | a → A |
| Translation | French to English | bonjour → hello |
| | Spanish to English | hola → hello |
| Linguistic | Present to gerund | go → going |
| | Singular to plural | cat → cats |
| | Antonyms | happy → sad |
| Knowledge | Country to Capital | France → Paris |
| | Person to Language | Macron → French |

Table 1: A representative subset of the tasks used in the study with input → output examples.

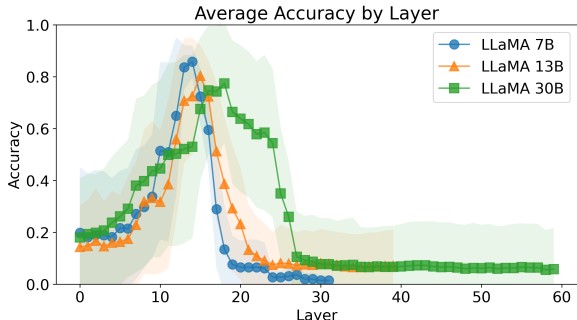

Figure 3: Accuracy for each choice of the intermediate layer $L$, averaged across all tasks. Solid lines show average values, and shaded areas standard deviations.

### 3.2 Tasks and Models

**Tasks** We consider a diverse set of 18 tasks across 4 categories: algorithmic, translation, linguistic, and factual knowledge. For simplicity, we limit ourselves to single-token outputs. A representative subset of the tasks is described in Tab. 1. A complete detailed table, as well as more information regarding the data, are provided in § A.1.

**Models** We use multiple open LLMs: LLaMA 7B, 13B, and 30B (Touvron et al., 2023), GPT-J 6B (Wang and Komatsuzaki, 2021), and Pythia 2.8B, 6.9B, and 12B (Biderman et al., 2023).

### 3.3 Finding $L$

The mechanism we described in §2.2 has a free parameter - the layer $L$ where $\mathcal{A}$ ends and $f$ begins. We use the proposed $(\mathcal{A}, f)$ implementation for different choices of $L$ and evaluate the accuracy on a development set to find the best layer.

Fig. 3 shows the accuracy on the development set, for different choices of $L$. We focus here on the LLaMA models and include the rest in § A.2. Interestingly, all models exhibit a performance peak at a similar intermediate layer, irrespective of their parameters and layer count differences.

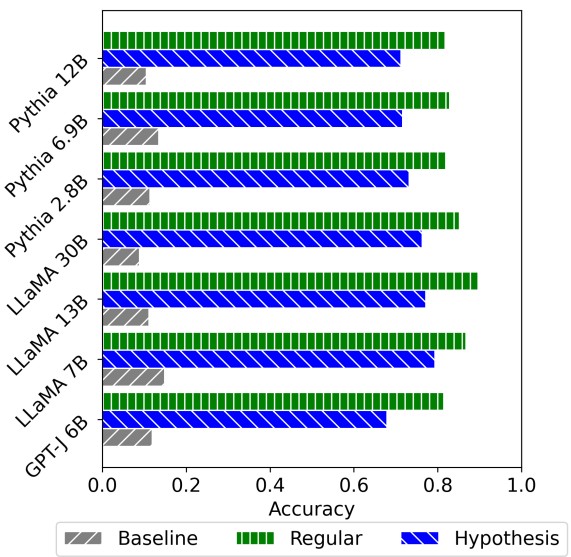

Figure 4: Average accuracy across all tasks for each model, using each of the three procedures: Baseline, Regular and Hypothesis.

## 3.4 Accuracy of Hypothesis Based Prediction

We next compare the accuracy of the $(\mathcal{A}, f)$ mechanism to that of a regular forward pass performing ICL. For each model and task, we evaluate the following three procedures:

- **Regular** An application of the LLM to the demonstrations $S$ and query $x$. Namely $T([S, x])$, as in regular ICL.

- **Hypothesis** Our proposed procedure from § 3.1 where $\mathcal{A}$ generates $\boldsymbol{\theta}$ using a dummy $x'$, and $f(\cdot; \boldsymbol{\theta})$ is applied to $x$ by running the transformer on $[x, \rightarrow]$ with $\boldsymbol{\theta}$ patched at layer $L$ of $\rightarrow$.

- **Baseline** A forward pass of the LLM only on $x$, without demonstrations $S$. That is, $T([x, \rightarrow])$. This is the same as the application of $f$ from our separated procedure, but without patching $\boldsymbol{\theta}$.

Fig. 4 shows the average accuracy across all tasks of these 3 procedures, for each model. Full results are reported in Tab. 6 in § A.2. Across all models, our procedure maintains around 80-90% of the accuracy of regular ICL, while the baseline reaches only 10-20%. This shows that our proposed separation to $\mathcal{A}$ and $f$ provides a good empirical approximation of the process underlying ICL.

## 4 Robustness of Task Vectors

In our setting, $\boldsymbol{\theta}$ is derived from $S$ and a dummy query $x'$. It is natural to examine the robustness of $\boldsymbol{\theta}$ to variations in these inputs. Intuitively, if it represents the task, it should remain stable across different $S$ and $x'$ values.

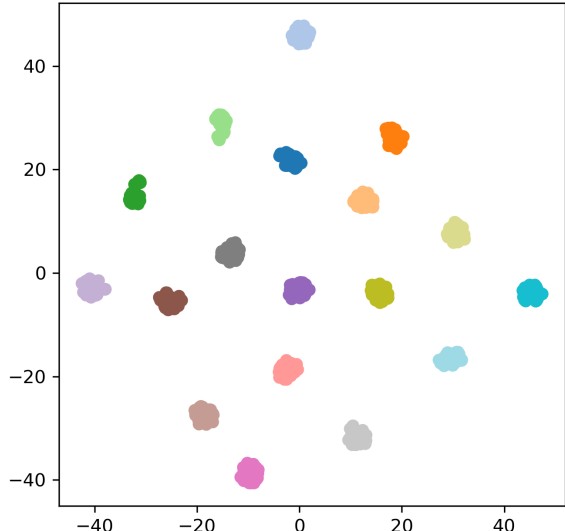

Figure 5: **A t-SNE plot of task vectors.** A 2D t-SNE plot visualizing 50 task vectors for each task, each generated from a different choice of $S$ and $x'$ using LLaMA 7B. Points are color-coded according to the task. Each task can be seen to form its own distinct cluster.

To test this, we use LLaMA 7B to generate 50 task vectors per task with varied $S$ and $x'$ and conduct two analyses.

**Geometry of $\boldsymbol{\theta}$** A t-SNE dimensionality reduction (Fig. 5) reveals that the task vectors form distinct clusters, each containing task vectors of a single task. Fig. 9 further shows proximity between tasks of the same category, strengthening the idea that they encapsulate task understanding.

**Variability of $\boldsymbol{\theta}$** Fig. 8 shows histograms of distances within and across tasks. It can be seen that vectors within the same task are closer than those between different tasks, indicating that $\boldsymbol{\theta}$ is stable within tasks and not highly influenced by $x'$ or $S$.

## 5 Dominance of $\boldsymbol{\theta}$ Patching

In §3 we prevented $f$ from directly accessing $S$. However, in a regular forward pass during ICL, the last token can attend to $S$. Here we verify that even in this case, $f$ mainly uses the task vector $\boldsymbol{\theta}$, without directly accessing the demonstrations $S$. To this end, we use a pair of tasks, $A$ and $B$, sharing the input space but differing on the output. We first use a "Regular" forward pass, where we provide the model with demonstrations $S$ for task $A$ (denoted $S_A$), to verify the model can perform this task using ICL. Then, we do a "Conflicting" forward pass, still providing $S_A$, while injecting $\boldsymbol{\theta}_B$. For more details, refer to Fig. 6 in §A.1.

| Task $A$ ($S$) | Task $B$ ($\theta$) | Regular Task $A$ | Conflicting Task $B$ |
|---|---|---|---|
| Next Letter | To Upper | 0.92 | 0.77 |
| List Last | List First | 0.95 | 0.78 |
| Present to Past | to Gerund | 0.96 | 0.95 |

Table 2: **Conflicting tasks experiment results.** The model's accuracy on the relevant task ($A$ in "Regular" and $B$ in "Conflicting") is displayed for both scenarios.

In Tab. 2, the "Regular" forward pass shows high accuracy on task $A$ (90%+), as anticipated. However, the "Conflicting" forward pass yields high accuracy on task $B$, corresponding to the injected task vector $\theta$. This implies that the model mainly relies on $\theta$, largely disregarding the demonstrations $S$ for task $A$. We note that the accuracy on task $B$ is slightly low, likely consistent with the performance dip seen in Fig. 6, and potentially further affected by the presence of $S$.

## 6 Interpreting $\theta$

The learned vector $\theta$ intuitively captures information about the task demonstrated by $S$. Here we provide evidence supporting this interpretation. Since $\theta$ is an intermediate hidden state of the transformer, we can employ a vocabulary projection method (nostalgebraist, 2020; Dar et al., 2022). Namely, we examine the top tokens in the distribution over the vocabulary induced by the hidden state.

Tab. 3 shows the top tokens for three tasks for LLaMA 13B (more models and tasks are provided in Tab. 7 in §A). In multiple cases, we observe tokens that directly describe the task. Importantly, these terms never explicitly appeared in the context. For example in the task of translation from French to English, we observe tokens such as "English" and "translate". This supports our view that $\theta$ carries significant, non-trivial semantic information about the task.

## 7 Related Work

**Emergence of ICL**   A key question with ICL is how it emerges as a capability from pre-training the LLMs. Levine et al. (2022) provides results in this direction that highlight the importance of training data structure. Xie et al. use probabilistic analysis and model pre-training data using Hidden Markov Models to theoretically explain the emergence of ICL, while Chan et al. (2022) empirically explore the effect of several distributional properties of the pre-training data.

| Task | Top tokens in the task vector projection |
|---|---|
| Previous Letter | e, y, unknown, alphabet, preceding, c Cad, zA, dit, bill |
| FR-EN | Mason, gram, immer, Santi, latin, utter, Span, Conc, English, equivalent |
| Present Simple to Gerund | cin, thats, gram, Lorenzo, cian, Isabel, uld, berto, partici, Sah |
| Country Capital | Paris, its, capital, central, Conc, cities, administrative, Los, Madrid, London |

Table 3: The top 10 tokens in the distribution induced by the task vector, for one task per category.

**Meta-Learning in Transformers**   Studies by Akyürek et al. (2022); von Oswald et al. (2022); Garg et al. focus on the meta-learning capabilities of transformers. They typically train models from scratch on elementary tasks such as linear regression, drawing theoretical parallels with algorithms like Gradient Descent and demonstrating how transformers could implement them. A key assumption of these works is a known parameter space within which gradient descent operates. Our work focuses on identifying such a parameter space for LLMs.

**ICL in LLMs**   Olsson et al. (2022) identify "induction heads" in transformers as a likely main mechanism of ICL. Dai et al. (2022) provide empirical evidence for the connection of ICL to Gradient Descent in LLMs, focusing on classification tasks. Concurrent work by Merullo et al. (2023) also explores a phenomenon similar to the task vectors we study here, where a single vector can encode learned functions. Our findings are complementary to theirs, and future work could explore the relationship between the two more closely.

## 8 Conclusions

Through this exploration of ICL in LLMs, we have shed light on a new perspective of ICL learning mechanisms. We have revealed a simple and elegant structure: ICL functions by compressing a given training set into a single task vector, which then guides the transformer to generate appropriate outputs given queries. Our work provides a stepping stone towards understanding how LLMs perform ICL. In light of our findings, future work could focus on understanding how the task vector is constructed as well as how it is used to calculate the output.

## Limitations

We study relatively simple tasks, whereas ICL can learn to perform more complex tasks, such as solving arithmetic reasoning problems. It remains to be seen if and how the mechanisms we observe here will translate to these cases. E.g., our approach focuses on cases where a single task vector suffices, while more complex ICL cases may require more elaborate parameterization. We also focus on tasks where the output is a single token, while some other tasks require multi-token outputs.

Finally, as noted above, we do not provide a mechanistic explanation for how the task vector is formed or how it is used. Namely, we do not explain how the transformer performs these calculations using its parameters.

## Acknowledgements

This project is funded by the European Research Council (ERC) under the European Unions Horizon 2020 research and innovation program (grant ERC HOLI 819080).

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

# A Appendix

Here we provide additional details and results.

## A.1 Additional Details

**Full Task Descriptions** Our study covers 18 tasks in 4 categories: Algorithmic, Translation, Linguistic and Knowledge. A detailed description of all tasks is provided in Tab. 5.

**Model Details** More details on the models used in the study are provided in Tab. 4.

**Task Data** Here we detail the sources of the data for each task. The accompanying GitHub repository contains the data itself as well as the code used to create it.

- Algorithmic: Generated programatically.

- Translation: For each language pair, the most frequent words in the source language are first retrieved from https://github.com/frekwencja/most-common-words-multilingual and are then translated to the destination language using the open-source package nltk.

- Linguistic: The data for the tenses tasks is parsed from https://github.com/Drulac/English-Verbs-Conjugates. The data for the plural-singular task is taken from https://github.com/sindresorhus/irregular-plurals. Finally, the data for the antonyms task is taken from https://github.com/SuzanaK/english_synonyms_antonyms_list.

- Knowledge Data for the knowledge tasks is taken from the counterfactual dataset introduced in (Meng et al., 2022).

**Conflicting Tasks Experiment** In Fig. 6, we provide more details and a visualization of the experiment described in §5.

## A.2 Additional Results

**Finding $\mathcal{A}$ and $f$** Fig. 7 shows results similar to Fig. 3, but for different models. It is interesting to observe that the curves are similar across different-sized models.

**Detailed results for Fig. 4.** Fig. 4 presented results for our $(\mathcal{A}, f)$ hypothesis-based approach, averaged across tasks. Table. 6 provides these results for all the specific tasks considered.

**Dependence of $\mathcal{A}$ on $x$** Fig. 9 and Fig. 8 provide more results on the geometry of the $\boldsymbol{\theta}$ vectors (see main text for discussion).

**Inspecting Task Vectors** Tab. 7 is an expanded version of Tab. 3, providing more vocabulary projections of $\boldsymbol{\theta}$ for additional tasks and on multiple LLMs.

| Model | Parameters | Dimension | Layers | Heads |
|-------|-----------|-----------|--------|-------|
| LLaMA | 7B | 4096 | 32 | 32 |
|       | 13B | 5120 | 40 | 40 |
|       | 30B | 6656 | 60 | 52 |
| GPT-J | 6B | 4096 | 28 | 16 |
| Pythia | 2.8B | 2560 | 32 | 32 |
|        | 6.9B | 4096 | 32 | 32 |
|        | 12B | 5120 | 36 | 40 |

Table 4: The models used in the study, with architectural information.

| Category | Task | Description | Example |
|---|---|---|---|
| Algorithmic | List first | Given a list of letters, output the first letter | a,b,c → a |
| | List last | Given a list of letters, output the last letter | a,b,c → c |
| | Next letter | Given a letter in the English alphabet, output the next letter | a → b |
| | Previous letter | Given a letter in the English alphabet, output the previous letter | b → a |
| | To lowercase | Given an uppercase letter, output the corresponding lowercase letter | A → a |
| | To uppercase | Given a lowercase letter, output the corresponding uppercase letter | a → A |
| Translation | French to English | Given a word in French, translate to English | bonjour → hello |
| | Spanish to English | Given a word in Spanish, translate to English | hola → hello |
| | English to Spanish | Given a word in English, translate to Spanish | hola → hello |
| | English to Spanish | Given a word in English, translate to French | hola → hello |
| Linguistic | Present to gerund | given an English verb in present simple tense, output the corresponding gerund form | go → going |
| | Present to past | given an English verb in present simple tense, output the corresponding verb in past simple | go → went |
| | Singular to plural | Given an English noun in singular form, output the plural form | cat → cats |
| | Antonyms | Given an English adjective, output an antonym | happy → sad |
| Knowledge | Country to Capital | Given a name of a country, output the name of the capital city | France → Paris |
| | Person to Language | Given a name of a person, output their native language | Macron → French |
| | Location to Continent | Given a name of a person, output their native language | Paris → Europe |
| | Religion | Given a name of a location or a person, output the associated religion | Muhammad → Islam |

Table 5: The tasks used in the study with input → output examples.

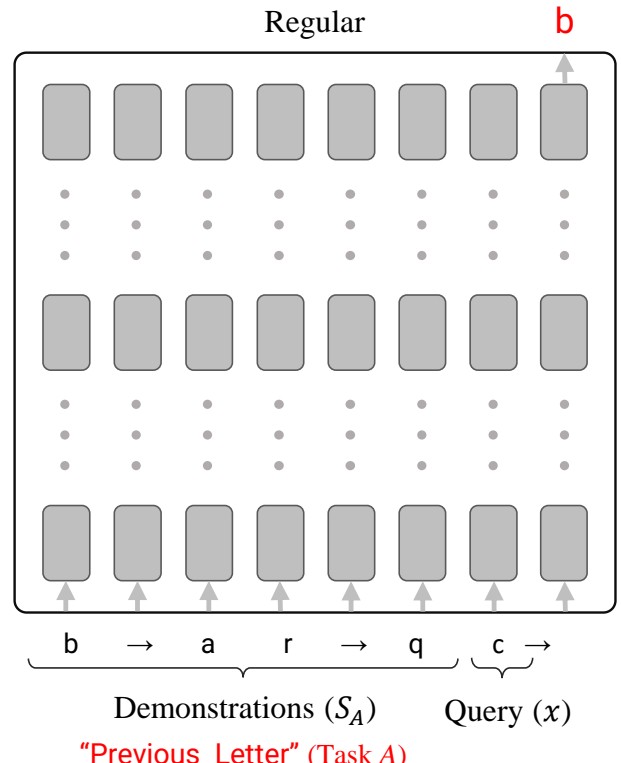

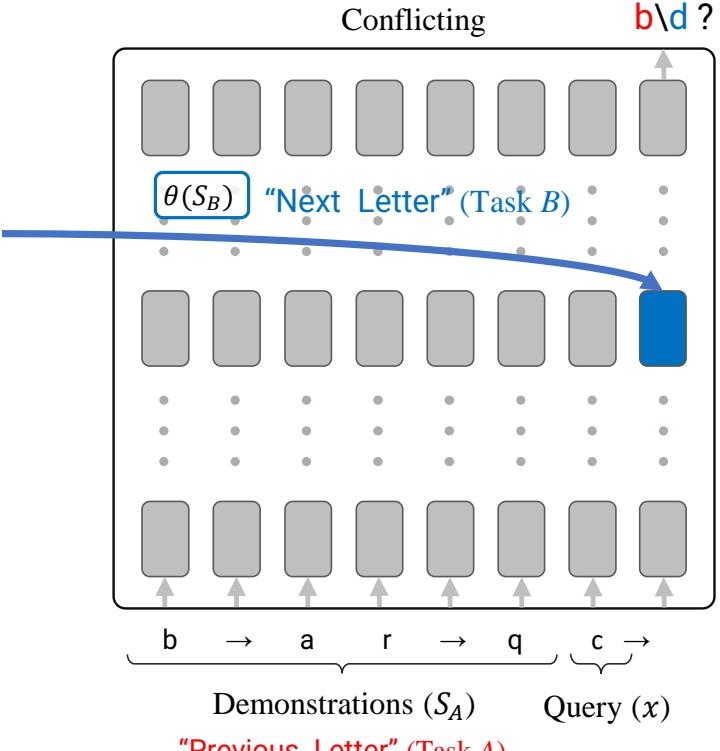

Figure 6: **Conflicting tasks experiment.** In the "**Regular**" scenario (top), the model is simply provided with demonstrations $S_A$ for Task $A$ (e.g. outputting the previous letter in the alphabet). In the "**Conflicting**" scenario (bottom), the model is still provided with demonstrations for Task $A$, but we inject a task vector $\boldsymbol{\theta}(S_B)$ from a conflicting Task $B$ (e.g. outputting the next letter in the alphabet).

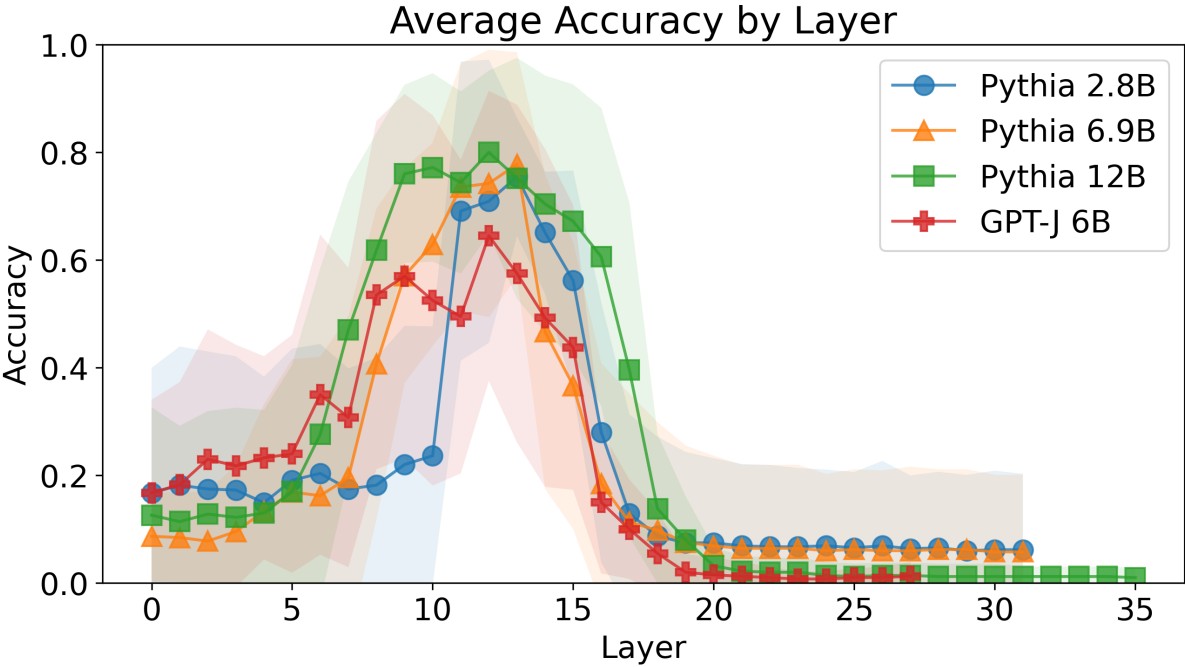

Figure 7: Accuracy for each choice of $L$ (the intermediate layer where the task vector is injected), averaged across all tasks. The solid line represents the average value, and the shaded area depicts the standard deviation.

Table 6: Complete results for Figure 4, reported for all tasks and models.

| Model | Task type | method
Task name | Baseline | Hypothesis | Regular |
|---|---|---|---|---|---|
| GPT-J 6B | Algorithmic | List first | 0.30 | 0.74 | 0.98 |
| | | List last | 0.24 | 0.64 | 1.00 |
| | | Next letter | 0.16 | 1.00 | 0.86 |
| | | Prev letter | 0.10 | 0.36 | 0.42 |
| | | To lower | 0.00 | 0.46 | 1.00 |
| | | To upper | 0.00 | 0.94 | 1.00 |
| | Knowledge | Country capital | 0.19 | 0.72 | 0.80 |
| | | Location continent | 0.03 | 0.58 | 0.70 |
| | | Location religion | 0.09 | 0.68 | 0.78 |
| | | Person language | 0.02 | 0.82 | 0.82 |
| | Linguistic | Antonyms | 0.43 | 0.68 | 0.78 |
| | | Plural singular | 0.08 | 0.90 | 0.98 |
| | | Present simple gerund | 0.00 | 0.88 | 0.98 |
| | | Present simple past simple | 0.02 | 0.76 | 0.96 |
| | Translation | En es | 0.14 | 0.34 | 0.56 |
| | | En fr | 0.16 | 0.36 | 0.54 |
| | | Es en | 0.06 | 0.70 | 0.74 |
| | | Fr en | 0.13 | 0.66 | 0.76 |
| LLaMA 13B | Algorithmic | List first | 0.77 | 1.00 | 1.00 |
| | | List last | 0.07 | 0.70 | 0.92 |
| | | Next letter | 0.31 | 1.00 | 0.94 |
| | | Prev letter | 0.05 | 0.34 | 0.50 |
| | | To lower | 0.00 | 0.94 | 1.00 |
| | | To upper | 0.00 | 0.94 | 1.00 |
| | Knowledge | Country capital | 0.17 | 0.84 | 0.86 |
| | | Location continent | 0.01 | 0.70 | 0.80 |
| | | Location religion | 0.10 | 0.74 | 0.84 |
| | | Person language | 0.02 | 0.76 | 0.88 |
| | Linguistic | Antonyms | 0.19 | 0.74 | 0.80 |
| | | Plural singular | 0.24 | 0.84 | 0.88 |
| | | Present simple gerund | 0.00 | 0.96 | 0.96 |
| | | Present simple past simple | 0.01 | 1.00 | 0.98 |
| | Translation | En es | 0.05 | 0.78 | 0.82 |
| | | En fr | 0.15 | 0.70 | 0.84 |
| | | Es en | 0.29 | 0.76 | 0.88 |
| | | Fr en | 0.25 | 0.54 | 0.72 |
| LLaMA 30B | Algorithmic | List first | 0.96 | 0.98 | 1.00 |
| | | List last | 0.02 | 0.64 | 0.96 |
| | | Next letter | 0.30 | 0.98 | 0.96 |
| | | Prev letter | 0.02 | 0.56 | 0.80 |
| | | To lower | 0.00 | 1.00 | 1.00 |
| | | To upper | 0.00 | 0.90 | 1.00 |
| | Knowledge | Country capital | 0.27 | 0.72 | 0.88 |
| | | Location continent | 0.01 | 0.70 | 0.86 |
| | | Location religion | 0.05 | 0.70 | 0.88 |
| | | Person language | 0.01 | 0.72 | 0.90 |
| | Linguistic | Antonyms | 0.37 | 0.76 | 0.82 |
| | | Plural singular | 0.21 | 0.84 | 0.90 |
| | | Present simple gerund | 0.00 | 0.76 | 0.98 |
| | | Present simple past simple | 0.02 | 0.98 | 1.00 |
| | Translation | En es | 0.07 | 0.74 | 0.78 |
| | | En fr | 0.10 | 0.80 | 0.86 |
| | | Es en | 0.24 | 0.70 | 0.88 |
| | | Fr en | 0.20 | 0.62 | 0.78 |
| LLaMA 7B | Algorithmic | List first | 0.87 | 0.98 | 1.00 |
| | | List last | 0.03 | 1.00 | 1.00 |
| | | Next letter | 0.03 | 0.94 | 0.88 |
| | | Prev letter | 0.04 | 0.52 | 0.58 |
| | | To lower | 0.00 | 0.74 | 1.00 |
| | | To upper | 0.00 | 0.60 | 1.00 |
| | Knowledge | Country capital | 0.28 | 0.82 | 0.86 |
| | | Location continent | 0.02 | 0.68 | 0.72 |
| | | Location religion | 0.12 | 0.84 | 0.94 |
| | | Person language | 0.02 | 0.68 | 0.78 |
| | Linguistic | Antonyms | 0.33 | 0.74 | 0.76 |
| | | Plural singular | 0.15 | 0.84 | 0.88 |

| Model | Task type | method Task name | Baseline | Hypothesis | Regular |
|---|---|---|---|---|---|
| | | Present simple gerund | 0.00 | 0.74 | 0.90 |
| | | Present simple past simple | 0.02 | 0.94 | 0.92 |
| | Translation | En es | 0.07 | 0.78 | 0.76 |
| | | En fr | 0.04 | 0.78 | 0.88 |
| | | Es en | 0.21 | 0.68 | 0.92 |
| | | Fr en | 0.15 | 0.66 | 0.70 |
| Pythia 12B | Algorithmic | List first | 0.53 | 0.98 | 0.96 |
| | | List last | 0.09 | 0.98 | 1.00 |
| | | Next letter | 0.15 | 0.96 | 0.76 |
| | | Prev letter | 0.00 | 0.24 | 0.42 |
| | | To lower | 0.02 | 1.00 | 1.00 |
| | | To upper | 0.00 | 0.98 | 1.00 |
| | Knowledge | Country capital | 0.19 | 0.58 | 0.82 |
| | | Location continent | 0.01 | 0.68 | 0.80 |
| | | Location religion | 0.07 | 0.64 | 0.78 |
| | | Person language | 0.01 | 0.72 | 0.86 |
| | Linguistic | Antonyms | 0.34 | 0.72 | 0.74 |
| | | Plural singular | 0.18 | 0.80 | 0.84 |
| | | Present simple gerund | 0.00 | 0.86 | 0.96 |
| | | Present simple past simple | 0.01 | 0.76 | 0.94 |
| | Translation | En es | 0.10 | 0.44 | 0.72 |
| | | En fr | 0.16 | 0.48 | 0.54 |
| | | Es en | 0.05 | 0.68 | 0.80 |
| | | Fr en | 0.14 | 0.68 | 0.80 |
| Pythia 2.8B | Algorithmic | List first | 0.69 | 0.96 | 1.00 |
| | | List last | 0.06 | 0.98 | 1.00 |
| | | Next letter | 0.42 | 0.86 | 0.90 |
| | | Prev letter | 0.01 | 0.22 | 0.48 |
| | | To lower | 0.00 | 1.00 | 1.00 |
| | | To upper | 0.00 | 1.00 | 1.00 |
| | Knowledge | Country capital | 0.18 | 0.70 | 0.76 |
| | | Location continent | 0.01 | 0.62 | 0.72 |
| | | Location religion | 0.08 | 0.76 | 0.82 |
| | | Person language | 0.00 | 0.82 | 0.82 |
| | Linguistic | Antonyms | 0.37 | 0.68 | 0.76 |
| | | Plural singular | 0.13 | 0.70 | 0.78 |
| | | Present simple gerund | 0.00 | 0.86 | 0.96 |
| | | Present simple past simple | 0.03 | 0.80 | 0.92 |
| | Translation | En es | 0.10 | 0.26 | 0.76 |
| | | En fr | 0.16 | 0.28 | 0.60 |
| | | Es en | 0.08 | 0.76 | 0.82 |
| | | Fr en | 0.10 | 0.64 | 0.82 |
| Pythia 6.9B | Algorithmic | List first | 0.43 | 1.00 | 0.98 |
| | | List last | 0.08 | 0.60 | 0.98 |
| | | Next letter | 0.01 | 0.66 | 0.86 |
| | | Prev letter | 0.04 | 0.28 | 0.32 |
| | | To lower | 0.00 | 1.00 | 1.00 |
| | | To upper | 0.00 | 0.94 | 1.00 |
| | Knowledge | Country capital | 0.21 | 0.76 | 0.82 |
| | | Location continent | 0.01 | 0.62 | 0.78 |
| | | Location religion | 0.10 | 0.80 | 0.80 |
| | | Person language | 0.01 | 0.76 | 0.80 |
| | Linguistic | Antonyms | 0.33 | 0.72 | 0.74 |
| | | Plural singular | 0.14 | 0.78 | 0.88 |
| | | Present simple gerund | 0.00 | 0.82 | 0.94 |
| | | Present simple past simple | 0.02 | 0.88 | 0.96 |
| | Translation | En es | 0.11 | 0.46 | 0.70 |
| | | En fr | 0.21 | 0.36 | 0.60 |
| | | Es en | 0.06 | 0.72 | 0.82 |
| | | Fr en | 0.14 | 0.66 | 0.74 |

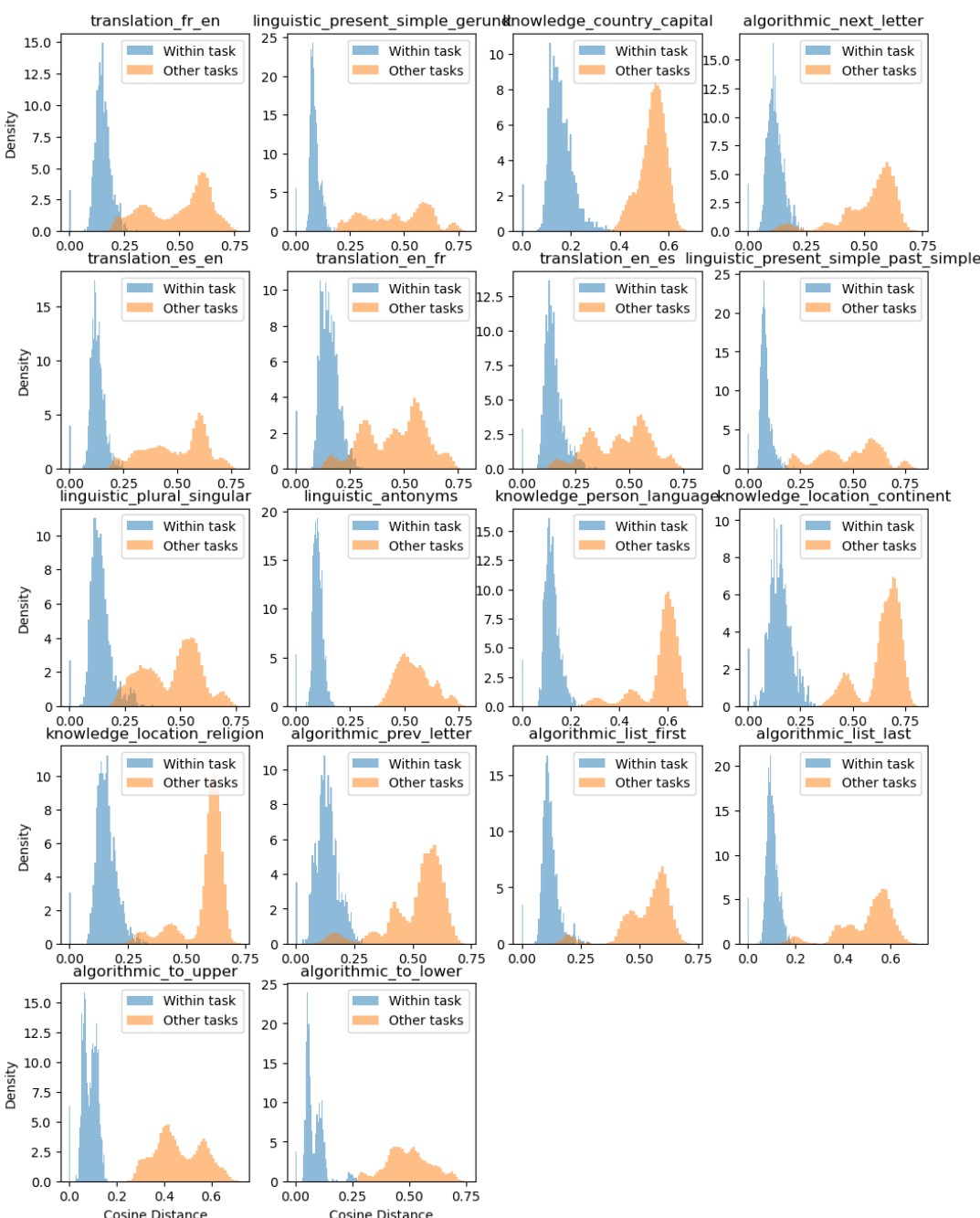

Figure 8: **Task Vector Variability**. For each task, two histograms are shown: (blue) the distribution of distances between different task vectors of this task, created from different $S$ and $x'$; (orange) the distribution of distances between task vectors of the task and of other tasks.

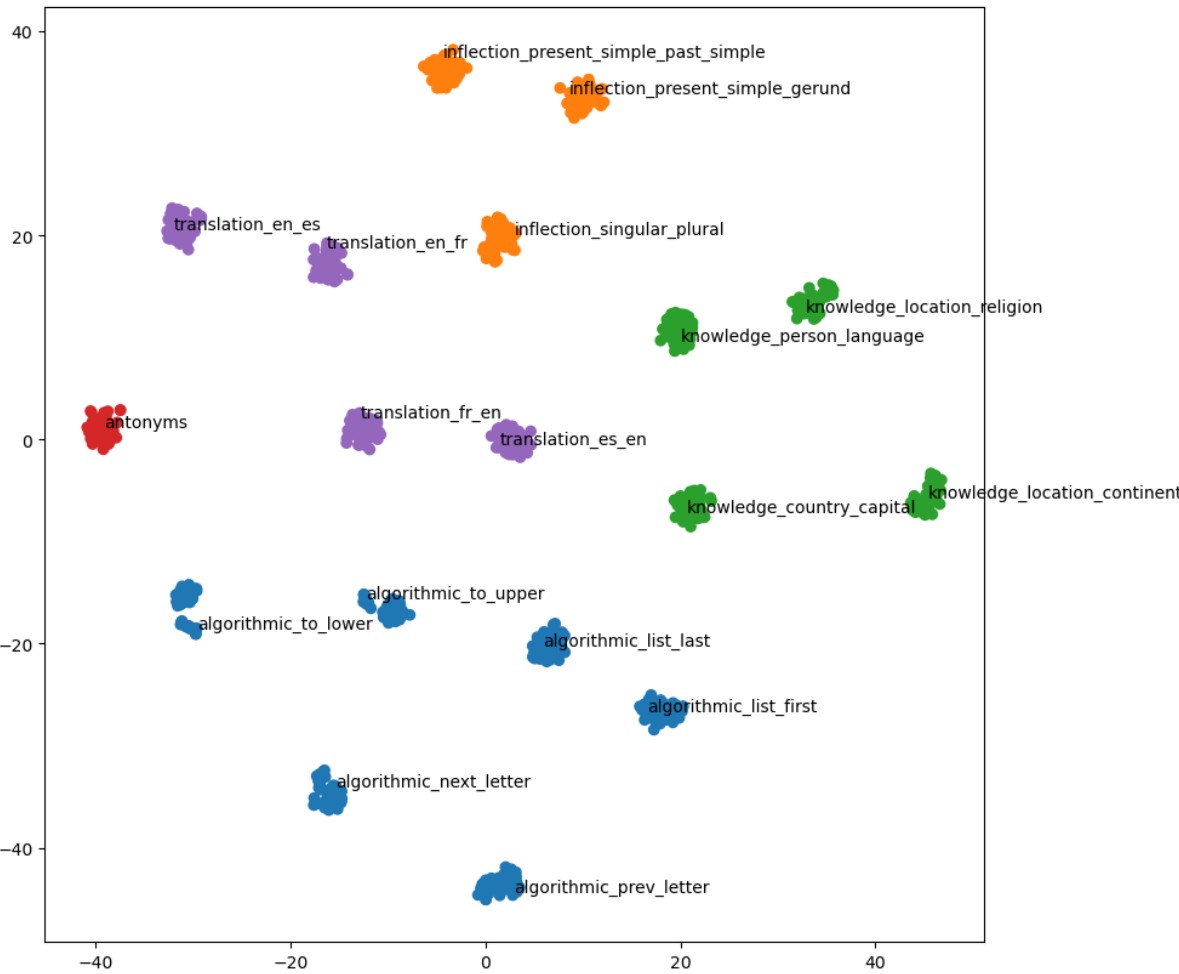

Figure 9: A 2D t-SNE plot, visualizing 50 task vectors for each task, each generated from a different choice of $S$ and $x$ using LLaMA 7B. Points are color-coded according to task category, such as algorithmic or translation. Each task can be seen to form its own distinct cluster. The labels provide the full name of the task in the cluster.

| Model | Task | Tokens |
|---|---|---|
| LLaMA 13B | Prev Letter | e, y, unknown, alphabet, preceding, c, Cad, zA, dit, bill, closer, etc, Stuart, aa, null, cin, ads, g, ulo, Ku |
| | FR-EN | Mason, gram, immer, Santi, latin, utter, Span, Conc, English, equivalent, engl, Usage, none, pron, ulo, translate, adu, Wiel, grammar, ML |
| | Present Simple to Gerund | cin, thats, gram, Lorenzo, cian, Isabel, uld, berto, partici, Sah, reporting, eing, tc, Roberto, habit, Writing, etc, ientos, ores, Dutch |
| | Country Capital | Paris, its, capital, central, Conc, cities, administrative, Los, Madrid, London, San, Isabel, exec, Ar, Bel, Wars, name, capit, Battle, History |
| Pythia 12B | Prev Letter | r, b, a, d, m, e, p, n, t, u, h, f, c, in, g, s, the, ar, l, x |
| | FR-EN | in, and, m, d, a, or, out, the, t, o, so, c, con, have, act, e, s, is, all, to |
| | Present Simple to Gerund | in, t, m, r, a, and, the, ing, action, d, o, e, current, simple, te, w, not, have, out, what |
| | Country Capital | the, in, a, C, N, B, L, M, T, P, S, R, G, and, F, I, K, U, D, H |
| GPT-J 6B | Prev Letter | b, c, v, g, s, name, i, ro, n, j, d, t, A, ai, com, m, ust, test, active, k |
| | FR-EN | other, name, the, true, is, social, s, active, time, car, type, money, F, force, a, public, heart, one, ms, life |
| | Present Simple to Gerund | getting, storing, working, moving, playing, doing, making, driving, shooting, picking, being, sending, putting, selling, watching, changing, taking, collecting, feeding, reading |
| | Country Capital | London, Paris, New, West, Berlin, South, Tokyo, San, Chicago, City, Moscow, Jerusalem, Amsterdam, Philadelphia, East, Madrid, Vienna, Beijing, Mexico, Germany |

Table 7: The top 20 tokens in the distribution induced by the task vector, for one task per category.