# OpenReview forum: "In-Context Learning Creates Task Vectors"
_EMNLP/2023/Conference — EMNLP 2023 Findings_

### Official Review · Reviewer_qoQT · 2023-07-23

**Soundness:** 3

**Excitement:**

3: Ambivalent: It has merits (e.g., it reports state-of-the-art results, the idea is nice), but there are key weaknesses (e.g., it describes incremental work), and it can significantly benefit from another round of revision. However, I won't object to accepting it if my co-reviewers champion it.

**Paper Topic And Main Contributions:**

The paper interpreted the so-called in-context learning(ICL) as a mechanism which first compresses the training samples to a task vector and then the transformer guided by the task vector would act in the input variables. It gave extensive experiments to illustrate the idea hypothesis.

**Questions For The Authors:**

(1) How do interpret the performance degradation in the experiments?
(2) attention blocks unavoidably access the inputs x in the real inference, how to interpret this with the hypothesis of task vector, which is claimed independent of x?

**Reasons To Accept:**

it sheds some light on the working mechanism of in-context learning observed in LLMs. The numerical experiments gave a fairly well approximation.

**Reasons To Reject:**

The overall idea and experiments are quite simple, and the total contribution seems insufficient with limited knowledge gain and novelty for the community.  The numerical approximation gives a bit degraded performance compared with the ICL. More deep and technical insights would be helpful such as the analysis of attention mechanism behavior in the ICL setting.


**Reproducibility:**

2: Would be hard pressed to reproduce the results. The contribution depends on data that are simply not available outside the author's institution or consortium; not enough details are provided.

**Reviewer Confidence:**

5: Positive that my evaluation is correct. I read the paper very carefully and I am very familiar with related work.

---

> ### Author Rebuttal · Authors · 2023-08-28
>
> We thank the reviewer for thoroughly reading our work and for the great questions!
>
> We address the concerns under "reasons to reject":
>
> **(1) Regarding the simplicity and apparently limited novelty of our experiments.**
>
> While the idea and experiments may appear simple, we believe that the simplicity can be seen as a strength, fostering accessibility and further exploration within the community. More importantly, our observation, though simple to understand in retrospect, has not been previously recognized, and we consider it to be far from trivial. The model's internal computation could have been implemented in many ways, and the emergence and utilization of these intermediate task vectors is just one possible (relatively sparse) solution among many. Our findings introduce new avenues for research that focus on both the efficiency and understanding of ICL. To highlight the broader implications and emphasize the novelty, we will include a detailed discussion in the camera-ready version.
>
> **(2) On the slightly lower performance of our method compared to ICL.**
>
> You are correct in noticing the slightly degraded performance with our task vector method compared to ICL. While this discrepancy exists, it doesn't undermine our main claims. Rather, it suggests that there may be additional mechanisms at play or that our method might not be perfectly capturing every aspect. If enhancing performance is a priority, future work could build on our approach, possibly by averaging over more demonstrations. Our main contribution lies in the insights provided, not necessarily in achieving optimal performance.
>
> **(3) A request for more insights into the attention mechanism behavior in the ICL setting.**
>
> While we acknowledge the potential value of a deeper analysis of the attention mechanism behavior in the ICL setting, it's important to note that this investigation requires substantial effort and could constitute a separate study. We believe this line of inquiry, though intriguing, is outside the scope of our current work, and we plan to consider it for future research. Thank you for pointing out this interesting direction!
>
> We further answer the reviewer's questions:
>
> **Q1: On the slightly lower performance of our method compared to ICL.**
>
> Please refer to our response above to the "reasons to reject" (point 2).
>
> **Q2: Regarding the access to the query x during regular inference and how it relates to our claim that the task vector is independent of the query.**
>
> Thank you for raising this important question, as this is a central point in our work. The idea is that when creating a task vector, the query can be any valid input for the task, and not necessarily the query that the model will provide the output for. This is demonstrated by using a different x' to create a task vector and then using this task vector to get an output for x. That is, while initial attention layers indeed access the query during real inference, the created task vector is still general to the task and can work with any query. This mechanism lets the model understand the "type" of the input query rather than the specific query.
>
> **Regarding Reproducibility:** We wish to clarify that, upon acceptance, we plan to publicly release our code for generating the data for all tasks and for reproducing the experiments reported in the paper. Our code is in Python and relies on standard libraries and openly accessible models.

---

### Official Review · Reviewer_PY9i · 2023-07-27

**Soundness:** 3

**Excitement:**

4: Strong: This paper deepens the understanding of some phenomenon or lowers the barriers to an existing research direction.

**Paper Topic And Main Contributions:**

The paper provides a new perspective of In-Context Learning (ICL) mechanisms for LLMs. Particularly, the authors claim that ICL is able to compress a given training set into a single task vector, and it guides the transformer to generate outputs given some query. This work propose a view of hypothesis space that ICLs learn with and show the validity on LLMs. The paper supports this claim using adequate experiments across several LLM and multiple tasks. Their work might contribute to the community with a better understanding of how LLMs perform ICL.

**Questions For The Authors:**

1. Regarding Figure 2, is there any reasoning on the behavior that layers of 20-25 led to much worse accuracy for LLaMA 7B or 13B?
2. The recently work by Merullo et al. (2023) is closed related to this paper. Could the authors provide more insights on how these two views are under unified experimental scenarios (i.e. the same model and task)?

**Reasons To Accept:**

The paper provides a novel viewpoint to understand the mechanism of ICL for LLMs. The two key components, compressing S into a single task vector, and using this task vector to modulate a transformer, are well presented and validated. The paper was clearly written; explained the intuition well and provided solid experiments to support the claims.

**Reasons To Reject:**

I am looking for some theoretical analyses underlying such interpretation of ICL. Also, connections between the proposed viewpoint and other existing work (e.g. Merullo et al. (2023)) shall be more discussed.

**Reproducibility:**

4: Could mostly reproduce the results, but there may be some variation because of sample variance or minor variations in their interpretation of the protocol or method.

**Reviewer Confidence:**

3: Pretty sure, but there's a chance I missed something. Although I have a good feel for this area in general, I did not carefully check the paper's details, e.g., the math, experimental design, or novelty.

**Typos Grammar Style And Presentation Improvements:**

Minor: Line 45-46 "then the class f would be linear classifiers..." a linear classifier is just an example and apparently it is not necessarily the case. Perhaps change "would" to "could"?

---

> ### Author Rebuttal · Authors · 2023-08-28
>
> We appreciate the reviewer's thoughtful examination of our work and the positive response! We are encouraged by the recognition of our contribution, clarity, and solid experimentation.
>
> We address the concerns under "reasons to reject":
>
> **(1) Regarding theoretical analyses underlying such interpretation of ICL.**
>
> While we agree that theoretical insights can provide a foundational understanding, we believe that empirical observations play a crucial role, particularly when exploring complex phenomena where theoretical models may be intractable or incomplete. Our empirical evidence not only sheds light on specific mechanisms but also paves the way for new theoretical directions. Specifically, we believe that the notion of a hypothesis class is key to any theoretical analysis of learning and in particular ICL. For example, previous theoretical approaches to ICL considered gradient descent as a possible mechanism for ICL (e.g. [1] and [2]). However, these works considered a synthetic setting where the hypothesis space was simple and clear. Here we provide an important step towards what could be a realistic hypothesis class. This can result in many exciting follow-ups that study both the algorithmic structure of ICL (i.e., how ICL finds a good element in the hypothesis class), as well as generalization bounds (that could, for example, rely on the low-dimensional representation of the task vectors).
>
> **(2) A request for more discussion on the connections between the proposed viewpoint and existing work (e.g. Merullo et al. (2023)).**
>
> Thank you for this comment. Note that specifically Merullo et al., 2023 was published within less than 3 months from the EMNLP 2023 submission deadline, so it is considered a concurrent work. Nonetheless, we agree that an extended discussion will be helpful.
>
> We propose a possible reconciliation of our work and Merullo et al., 2023. Considering the residual stream at the last position, the process might look as follows: first, the task vector is created from the demonstrations (a part which Merullo et al. don't discuss), then the query is integrated to the last position from which the prediction is obtained (as shown in their work), and finally, the mapping is applied (perhaps as they describe). Their discussion about the application of the mapping using an FFN mechanism might be seen as complementary to our work, as it provides an explanation for what we term f(x, θ). It's important to note, however, that they demonstrate this result only for GPT-2, which is a special case due to the weight tying between input and output embeddings. This may be why they observe a simple word2vec-like mechanism. Our results are more general as they hold across a wide range of models, some without such weight-tying. Their distinction between abstractive and extractive tasks is interesting but might not provide the complete picture, as we see our method works in the list first/last tasks, which are extractive tasks.
>
> We will include this discussion in the camera-ready version.
>
> We further answer the reviewer's questions:
>
> **Q1: Regarding the seeming qualitative difference between different models in Figure 2.**
>
> Thanks for this question. Please note that this decline in accuracy in upper layers also occurs in LLaMA 30B (and other models, please see Figure 4 in the appendix). In Figure 2 we just cut the x-axis earlier, for readability, which prevents seeing this decrease for LLaMA 30B (which has 60 layers, compared to only 30-40 in the smaller models). But given your question we realize that this might be confusing. To address this, we will replace Figure 2 with a version where the x-axis is not cut, in which it is possible to see that the accuracy for LLaMA 30B shows a similar decrease in upper layers. E.g. at around layer 30, the accuracy of LLaMA 30B drops to a value of around 0.1.
>
> Also, we will include another plot where the x-axis is normalized with respect to the total number of layers for each model – this plot (that we cannot share in this restricted format, we apologize again) shows that trends across models align more closely.
>
> **Q2: Regarding concurrent related work.**
>
> Please refer to our response above to the "reasons to reject" (point 2).
>
> **Regarding typos:** We will fix this, thanks!
>
> [1] Transformers learn in-context by gradient descent. von Oswald et al., ICML 2023.
>
> [2] Why Can GPT Learn In-Context? Language Models Implicitly Perform Gradient Descent as Meta-Optimizers. Dai et al., ACL 2023.

---

### Official Review · Reviewer_s9kR · 2023-08-05

**Soundness:** 3

**Excitement:**

3: Ambivalent: It has merits (e.g., it reports state-of-the-art results, the idea is nice), but there are key weaknesses (e.g., it describes incremental work), and it can significantly benefit from another round of revision. However, I won't object to accepting it if my co-reviewers champion it.

**Missing References:**

Potential missing reference: https://arxiv.org/abs/2205.05055.

**Paper Topic And Main Contributions:**

This paper attempts to formalize the notion of the hypothesis class of functions for ICL. The authors propose a novel approach to build task vectors from the training set, which is then used to answer queries. The authors conducted experiments using large language models such as GPT-J, Llama, and Pythia to support their findings. The results of these experiments demonstrate that the proposed approach is practical for a wide range of tasks. This work provides one possible way to interpret how LLMs operate internally for performing ICL.

**Questions For The Authors:**

- The exact mathematical formulation of f(x, \theta(S)) is unclear. Is it an inner product between two representations?
- Coming to the usage of x’ instead of x to get the representation, the model has seen the query x’. Will this not tamper the Vector for S( i.e training data for the ICL) as x’ does not have a token succeeding “->” ?
- Why do you think intermediate layers are more suitable as shown in the results in Figure 2? Also, is this pattern consistent across all subsets of tasks?
- What’s the variance of \thta(S) for different choices of x’? Is it very low? If that's not the case it will be hard to conclude that the transformer uses task vectors.
- Are there any insights based on results for different types of tasks?
- To elaborate, more experiments on validating whether tasks from a similar set  of tasks (Ex : En -> Fr, Fr -> En ) is nearer in n-d dimensional space rather than tasks that is dissimilar (Fruits -> Colour) would have made the analysis more stronger towards the positioning that ICL creates task vectors. This could have been validated with methods like t-sne etc.

**Reasons To Accept:**

The empirical insights gained from this study could be valuable to other researchers in the field who are working to understand the phenomenon of in-context learning.

**Reasons To Reject:**

- What’s the variance of \theta(S) for different choices of x’? Is it very low? If that's not the case it will be hard to conclude that the transformer uses task vectors. This is something where the empirical evidence provided by this work is not satisfactory.
- Also, the overall reasoning is somewhat weak. As even if the probe vectors show performance with a computation similar to the hypothesis class, it won't still be sufficient to imply that the hypothesis class is true of this particular type. Such empirical results in support of the claim are necessary. but not sufficient.

**Reproducibility:**

3: Could reproduce the results with some difficulty. The settings of parameters are underspecified or subjectively determined; the training/evaluation data are not widely available.

**Reviewer Confidence:**

3: Pretty sure, but there's a chance I missed something. Although I have a good feel for this area in general, I did not carefully check the paper's details, e.g., the math, experimental design, or novelty.

---

> ### Author Rebuttal · Authors · 2023-08-28
>
> We thank the reviewer for thoroughly reading our work and for the great questions!
>
> We address the concerns under "reasons to reject":
>
> **(1) Regarding the variance of θ(S).**
>
> Thanks for this great comment. We first explain why the variance between task vectors of the same task, obtained from different demonstrations S and query x', doesn't necessarily have to be low for our hypothesis-class view to hold. We then report new analysis results to show that, in practice, it is low.
>
> - The variance between task vectors of the same task does not necessarily have to be low: Note that for the hypothesis-class view to hold, we don't require that the thetas are close to each other in Euclidean space, but rather that they are "functionally equivalent" in the sense that they define the same function when used in f(x,θ). More formally, we can define the set of vectors Θ as all the θ that can be generated by different choices of x'. Each member of this set is a task vector, in the sense that it can be used with any query x to produce the correct output using f(x,θ). This phenomenon, where multiple parameter vectors (analogous to our task vectors) exist for any single dataset, can also happen in classical learning algorithms (e.g., there could be many hyperplanes that separate the same dataset). The above explanation shows that the variance does not necessarily have to be low in order for our hypothesis-class view to hold.
>
> - New results: Regardless of the above discussion, we agree with the reviewer that analyzing the variance of the task vector under different choices of x' (and S) is interesting, and we conducted an experiment following the reviewer's comment. We used LLaMA 7B and for each task, created a plot with two histograms, showing the cosine distances between (a) task vectors of the same task (with different S and x'), and (b) task vectors of different tasks. Across all the tasks, it is possible to see a clear separation between the two histograms, which provides evidence that the variance of task vectors of the same task is substantially smaller compared to between vectors of different tasks. Unfortunately, because of the restrictions on the rebuttal to include only text without links, we cannot share the plots. Therefore, we use the table below to report the mean and standard deviation of the aforementioned histograms, where the separation can be observed. We will include the full analysis in the revised manuscript.
>
> We hope that this clarified formulation and further analysis answers your concerns, and are happy to answer any further questions. We are also happy to provide the plots if the program chairs allow.
>
> The following table displays the new results. It includes the mean and standard deviation of the distances between task vectors within a task, as well as the distances to task vectors from other tasks. We can see that in all cases the in-task distances are much smaller compared to the distances to other tasks.
>
> | Task               |   Within Task Mean |   Within Task Std |   Between Tasks Mean |   Between Tasks Std |
> |:-------------------|---------------:|--------------:|--------------------:|-------------------:|
> | Next Letter        |          0.126 |         0.04  |               0.53  |              0.112 |
> | Prev Letter        |          0.128 |         0.048 |               0.557 |              0.111 |
> | List First         |          0.106 |         0.03  |               0.54  |              0.114 |
> | List Last          |          0.103 |         0.029 |               0.519 |              0.117 |
> | To Upper           |          0.084 |         0.029 |               0.464 |              0.102 |
> | To Lower           |          0.089 |         0.05  |               0.51  |              0.105 |
> | Fr En              |          0.159 |         0.038 |               0.5   |              0.12  |
> | Es En              |          0.115 |         0.034 |               0.524 |              0.086 |
> | En Fr              |          0.139 |         0.04  |               0.465 |              0.125 |
> | En Es              |          0.135 |         0.045 |               0.5   |              0.127 |
> | Gerund             |          0.087 |         0.023 |               0.48  |              0.145 |
> | Past Simple        |          0.094 |         0.026 |               0.465 |              0.138 |
> | Singular Plural    |          0.145 |         0.044 |               0.453 |              0.123 |
> | Antonyms           |          0.141 |         0.035 |               0.495 |              0.092 |
> | Country Capital    |          0.156 |         0.041 |               0.53  |              0.049 |
> | Person Language    |          0.13  |         0.037 |               0.57  |              0.085 |
> | Location Continent |          0.142 |         0.05  |               0.665 |              0.08  |
> | Location Religion  |          0.149 |         0.043 |               0.578 |              0.09  |
>
>
> **(2) Regarding the concern that "... As even if the probe vectors show performance with a computation similar to the hypothesis class, it won't still be sufficient to imply that the hypothesis class is true of this particular type. Such empirical results in support of the claim are necessary. but not sufficient."**
>
> Our view of a hypothesis class is a set of functions that any learned function will be a member of. In that sense, we do show that _the functions learned by ICL are in the class of transformers with task vectors as input_. We agree that we do not show mechanistically how these vectors are computed by the transformer (e.g. is there some implicit gradient descent implemented). However, we think that characterizing such a simple class of functions as the hypothesis class is an important step forward. We will make sure to clarify this.
>
> We further answer the reviewer's questions:
>
> **Q1: The exact mathematical formulation of f(x, θ(S)).**
>
> The notation f(x, θ(S)) does not refer to an inner product or specific mathematical operation. It simply symbolizes a computation f that relies on a query x and a task vector θ(S), without direct dependence on S.
>
> **Q2: The usage of x' instead of x to get the representation.**
>
> We appreciate the opportunity to clarify the role of x' in our framework. Note that x' is not part of the demonstrations set S, and its label is intentionally not provided. Rather, it functions as a query, keeping the structure of the prompt and signifying the type of input for which an output is expected. To elaborate, recall that the task vector is essentially an intermediate representation of the -\> token. To keep the structure of the prompt, the -\> needs to be preceded by an input, and we therefore put x' there. While in principle the task vector could be influenced by x', our empirical data, further supported by the above new histogram data (from the answer to concern (1)), demonstrate that its effect is weak.
>
> Essentially, x' helps in keeping the structure of the prompt for extracting the task vector from the demonstrations S. This task vector is then patched to the representation of x, which, until that point, was computed independently of both x' and S.
>
> We hope this explanation clears up any confusion. Please feel free to ask if you have further questions or require additional details.
>
> **Q3: The higher suitability of intermediate layers and the consistency of this pattern.**
>
> The pattern of intermediate layers being more suitable, as depicted in Figure 2, appears to be relatively consistent across the tasks we evaluated. Note that this is observable from the relatively low standard deviations shown in the figure, which were calculated across different tasks.
>
> The exact reason for this pattern is not fully clear. It might be that the model requires initial layers to process the input (i.e. represent the examples and identify the rules), and then utilize intermediate layers to aggregate this understanding to create the task vector at an intermediate layer. Further layers could apply the identified rule and prepare the output. This possible interpretation is in line with prior work such as [3] and [4]. However, this is a hypothesis, and additional work is needed to analyze the entire mechanism.
>
> **Q4: A suggestion for strengthening the analysis by validating whether similar tasks have similar task vectors, using methods like t-SNE.**
>
> We appreciate your recommendation to compare tasks in n-dimensional space using techniques like t-SNE. Following your suggestion, we conducted this analysis with t-SNE and found two reassuring phenomena: first, task vectors from the same task are closely clustered together, and second, similar tasks (from the same category) are indeed grouped together. Specifically, the inter-cluster standard deviation is 20 while the intra-cluster is ~1. These results are also consistent with the analysis of the histogram above (see our response regarding the variance of theta). Again, we are sincerely sorry that we cannot share the plot. Overall, this additional evidence supports our findings, and we will include it in the final version of the paper.
>
> **Regarding the missing reference:** Thanks, we will add this!
>
> **Regarding Reproducibility:** We wish to clarify that, upon acceptance, we plan to publicly release our code for generating the data for all tasks and for reproducing the experiments reported in the paper. Our code is in Python and relies on standard libraries and openly accessible models.
>
> [1] Interpretability in the Wild: a Circuit for Indirect Object Identification in GPT-2 small. Wang, et al., ICLR 2023.
>
> [2] The Hydra Effect: Emergent Self-repair in Language Model Computations. McGrath et al., 2023.
>
> [3] The Bottom-up Evolution of Representations in the Transformer: A Study with Machine Translation and Language Modeling Objectives. Voita et al., EMNLP 2019.
>
> [4] Dissecting Recall of Factual Associations in Auto-Regressive Language Models. Geva et al., 2023.

---

### Meta-Review · Area_Chair_8ZSY · 2023-09-15

**Recommendation:** 3

**Metareview:**

The paper delves into the mechanism of in-context learning (ICL) within LLMs and proposes a novel approach that involves compressing training samples into a task vector and using it to modulate a transformer for handling queries. Extensive experiments using various LLMs support this concept.

The reviewers collectively acknowledge the empirical insights gained from the study and agree on the potential value of this work for researchers seeking to understand ICL. Given the average scores being above the acceptance borderline and considering the valuable insights provided by the paper, the decision should lean towards acceptance. However, the reviewers highlight the need for further theoretical analyses, deeper insights, missing references and some presentation issues. It's recommended to encourage the authors to address some of these concerns in a revision as promised in the discussion, to strengthen the overall impact and quality of the paper.

---

### Decision · Program_Chairs · 2023-10-07

**Decision:**

Accept-Findings

**Comment:**

The paper delves into the mechanism of in-context learning (ICL) within LLMs and proposes a novel approach that involves compressing training samples into a task vector and using it to modulate a transformer for handling queries. Extensive experiments using various LLMs support this concept.

The reviewers collectively acknowledge the empirical insights gained from the study and agree on the potential value of this work for researchers seeking to understand ICL. Given the average scores being above the acceptance borderline and considering the valuable insights provided by the paper, the decision should lean towards acceptance. However, the reviewers highlight the need for further theoretical analyses, deeper insights, missing references and some presentation issues. It's recommended to encourage the authors to address some of these concerns in a revision as promised in the discussion, to strengthen the overall impact and quality of the paper.